# Ultralow Noise and Timing Jitter Semiconductor Quantum-Dot Passively Mode-Locked Laser for Terabit/s Optical Networks

Youxin Mao *, Guocheng Liu, Khan Zeb, Zhenguo Lu, Jiaren Liu, Philip J. Poole, Chun-Ying Song and Pedro Barrios

Advanced Electronics and Photonics Research Centre, National Research Council Canada, Ottawa, ON K1A 0R6, Canada
* Correspondence: youxin.mao@nrc-cnrc.gc.ca

**Abstract:** Diode optical frequency comb lasers are promising compact solutions to generate high-speed optical pulses for applications in high spectral efficiency wavelength division multiplexing transmission with advanced modulation formats. In this paper, an InAs/InP quantum dot (QDot) *C*-band single-section passively mode-locked laser (MLL) based broadband optical frequency comb source with a free spectral range of 28.4 GHz is presented. The device exhibits less than 1.5 MHz optical linewidth (phase noise) over 56 channels and 2.1 fs pulse-to-pulse timing jitter with a central wavelength of 1550 nm. Using this comb, we demonstrate an aggregate data transmission capacity of 12.5 Terabit/s over 100 km of standard single mode fiber by employing dual-polarization with 16 QAM modulation format. This investigation shows the viability for semiconductor QDot MLLs to be used as low-cost optical source in Terabit/s or higher optical networks.

**Keywords:** semiconductor quantum dot laser; passively mode-locked laser; InAs/InP quantum dot; wavelength-division multiplexing; quadrature amplitude modulation; Terabit/s optical networks





## 1. Introduction

Due to the ever-increasing internet traffic worldwide, it is expected that network communications capacities up to Terabit/s will be required where Gigabit/s speeds are already becoming standard [1]. Technical implementation of such networks requires advanced higher-order modulation formats along with parallel transmission on multitudes of wavelength channels to increase the spectral efficiency [2]. In these circumstances, optical frequency combs (OFCs) become particularly attractive light sources with large numbers of well-defined carriers for wavelength-division multiplexing (WDM) in a single semiconductor diode device. In particular, one of the important advantages of OFCs is that the comb lines are inherently equidistance in frequency, hence easing the requirements for inter-channel guard bands and avoiding frequency control of individual lines as needed in conventional schemes that combine arrays of independent *distributed-feedback* lasers.

OFCs generated by monolithic semiconductor mode-locked lasers (MLLs) have been demonstrated to be efficient for WDM based optical networks owing to compact size, low power consumption, wide optical bandwidth with a flat optical spectrum, and the ability for hybrid integration with silicon substrates. Monolithic MLLs have been widely studied in bulk and quantum well (QW) semiconductor devices for more than 30 years [3]. It has been demonstrated that using quantum dots (QDots) or quantum dashes (QDashes) as the active gain medium instead of QWs for semiconductor lasers can provide a number of enhancements in device performance [4]. This makes QDot or QDash single-section passively (SSP) MLLs very promising for creating OFCs for the next generation of high-speed networks [5]. The common implementation is a simple Fabry–Pérot (F–P) cavity laser which supports multiple longitudinal lasing modes with all the modes mutually phase locked [6]. Since there is no saturable absorber section, the SSP MLLs achieve

mode-locking results with an increased average output power, simpler fabrication, simpler operation, and the ability to operate at higher repetition rate [7]. Moreover, they offer reduced spontaneous emission rates and low threshold current densities, which leads to reduced intrinsic noise [8,9]. Although these lasers exhibit optical linewidths in the sub-MHz range, the radio frequency (RF) linewidth [10] and differential phase noise between adjacent comb lines is much narrower, showing high coherence among spectral lines thus enabling advanced modulation formats [11,12] and joint phase estimation [13].

Various publications have shown the potential of SSP QDot and QDash MLL for Terabit/s transmission [9,10,14–25], such as an aggregate capacity of 10 Terabit/s reported on 16 QAM dual-polarization 75 km standard single-mode fiber (SSMF) WDM transmission using 38 channels of a QDash-MLL [14]. It is also shown in [20] that by utilizing an external-cavity feedback technique, an increased aggregate capacity of 12 Terabit/s with 32 QAM WDM transmission could be achieved. However, external feedback makes the setup more complex, which may result in undesirable situations such as mode instability or hopping, that could reduce the bit error rate (BER) performance in data transmission [26]. In recent years, we have reported InAs/InP QDot and QDash MLLs operating in the *C*- and *L*-band with channel spacings from GHz to THz [10,18,23–25,27–32]. Comparison of Qdot vs. QDash lasers has suggested that QDots are favorable for mode-locking, as the improved height-diameter aspect provides a deeper confinement of the charge carriers [33]. QDot lasers have been shown to exhibit larger values of direct modulation bandwidth and relaxation oscillation frequency induced by the fast carrier capture from the wetting layer to the dots [34–36]. In our group we have also consistently observed shorter pulse durations with QDot based MLLs as compared to that of QDash and demonstrated it in high-capacity networks [32]. However, the systematic performance of QDot MLLs, especially for the timing jitter and noise mechanisms, have not been investigated. In this paper, we study detailed properties of a QDot based InAs/InP SSP MLL with a mode spacing of 28.4 GHz, including relative intensity noise (RIN), phase noise, pulse-to-pulse, and pulse-to-clock (integrated) timing jitter. Discussions of the comparison with published results are also presented. Moreover, the application of a 28.4 GHz InAs/InP QDot SSP MLL in a 12.5 Terabit/s optical network over 100 km of SSMF with 16 QAM modulation is experimentally demonstrated.

## 2. Materials and Methods

Figure 1a,b show a schematic cross-sectional diagram and scanning electron microscope (SEM) image of the InAs/InP F-P QDot SSP MLL with a single lateral mode ridge-waveguide structure. The laser structure was grown on a 3″ (001) oriented n-type InP substrate, with the n-type cladding and laser core grown by chemical beam epitaxy. An n-type InP cladding was first grown, followed by a 350 nm thick lattice matched InGaAsP waveguide core. The composition of InGaAsP was chosen to have a photoluminescence peak of 1.15 μm at 300 K. This core structure provides both carrier and optical confinement. By choosing appropriated growth conditions we could create either QDot or QDash layers as the gain medium [37] in the waveguide core. For this device, we grew five stacked layers of QDots in the center, forming the laser active region. A top view SEM image of typical surface InAs Qdot is shown Figure 1c. The five InAs dot layers were deposited, followed by a 25 s growth interruption for each layer to allow the In to diffuse on the surface and form dots. Each dot layer is then capped with a thin InP barrier layer and finally capped with the InGaAsP barrier material. This double-cap process was used to precisely control the emission wavelength [29]. The average QDot density in each active layer is around $3.5 \times 10^{10}$ cm$^{-2}$. An upper p-type InP cladding layer (containing an etch stop for ridge fabrication) and a heavily doped p-type InGaAs contact layer are then grown with metal organic chemical vapor deposition.

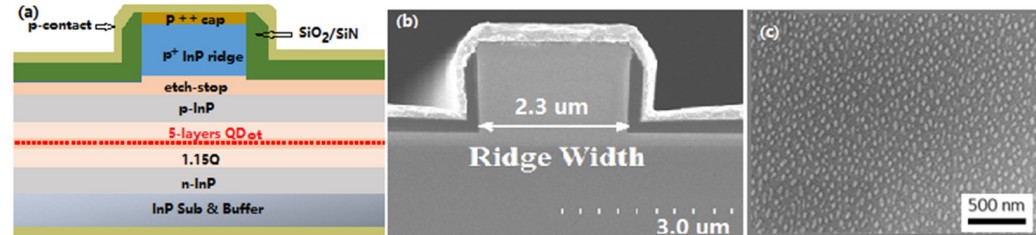

**Figure 1.** (**a**) Schematic cross-sectional diagram of the ridge-waveguide InAs/InP QDot laser structure; (**b**) SEM images of the cross-sectional facet of a fabricated F-P InAs/InP QDot laser; and (**c**) the top view of an InAs QDot layer.

Following growth, lasers are processed by defining ridges using a combination of dry etching into the InP cladding followed by wet etching to an etch stop layer as shown in Figure 1a. Finally, a SiO2/SiN isolation layer and a metal contact layer are deposited to form a low resistance Ohmic contact. Devices are then cleaved to cavity lengths of 1500 µm to form F-P lasers with a corresponding to 28.4 GHz mode spacing (repetition rate). A cleaved cross-section SEM image is shown in Figure 1b showing a waveguide width of 2.3 µm. No coatings were deposited on the cleaved facets.

For the experimental characterization the laser chip is mounted onto a commercially available aluminum nitride carrier to provide mechanical support with two gold electroplated pads to provide electrical connection to the laser chip. The laser chip is bonded substrate side down to one pad using eutectic gold tin, and the top contact is made through wire-bonding to the other pad. To reduce temperature fluctuations, this chip-on-carrier is placed on a copper block with a thermoelectric cooler underneath to maintain an operating temperature range of 18–20 °C. An ultra-low-noise battery powered laser diode driver controller (ILX Lightwave, Model LDC-3722) is used to DC bias the laser. The laser output light is collected from its front facet using a lensed polarization maintaining (PM) fiber attached to a two-stage PM isolator for reducing any back-reflections. The position of this lensed fiber is precisely adjustable in three dimensions for coupling the light optimally from the laser cavity. This laser is free run without any form of feedback to control the laser linewidth.

## 3. Experimental Results

### 3.1. Optical Spectrum and Light Current Characteristics

Figure 2a shows the lasing spectrum at a drive current of 400 mA, and Figure 2b single-mode fiber coupled light output of the laser as a function of drive current. During measurements, a heatsink temperature of 18 °C was used. The 3 dB bandwidth of the laser is 9.3 nm, from 1545.8 nm to 1555.2 nm, at a central wavelength of 1550.5 nm with 42 lasing modes. The 10 dB bandwidth is 12.5 nm, from 1543.3 nm to 1555.8 nm, at a central wavelength of 1549.6 nm with 56 lasing modes as indicated in the Figure 2a. For this device, an optical signal-to-noise ratio (OSNR) greater than 60 dB is achieved. Moreover, from Figure 2b, it can be seen that a threshold current of 60 mA and fiber coupled optical power of 24.4 mW at 450 mA are obtained.

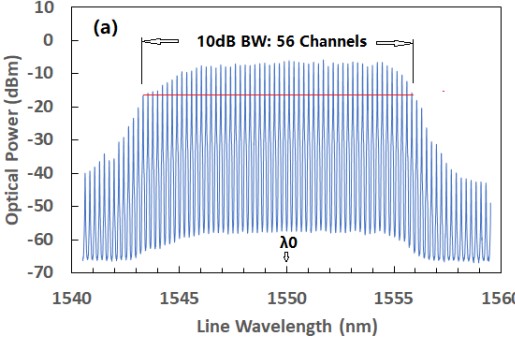
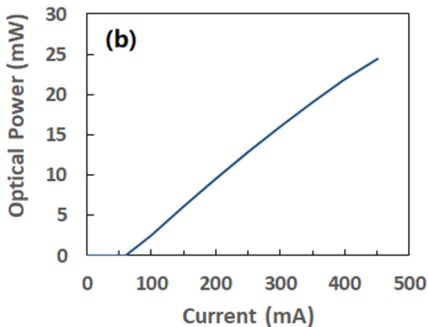

**Figure 2.** (**a**) Optical spectrum and (**b**) light-current characteristics of the single-mode fiber coupled laser measured at 400 mA and a temperature of 18 °C.

### 3.2. RIN

To investigate the RIN for the laser, the RIN spectra including all modes and individual longitudinal modes are measured in the frequency range from 10 MHz to 20 GHz using the Keysight N4371A RIN measurement system; examples are shown in Figure 3a. The RIN values from the entire spectral emission are all below –160 dB/Hz over the 20 GHz frequency range, and an integrated average RIN as low as −164.4 dB/Hz is achieved. The RIN spectra of individual longitudinal modes are assessed by selecting each mode using a tunable optical narrow bandwidth filter. Figure 3a shows a comparison of the RIN spectra of three modes at low (1546.04 nm), center (1550.58 nm), and long (1554.46 nm) wavelength. They all demonstrate similar behaviors and the RIN values measured at high frequencies (>4 GHz) drop down to below −140 dB/Hz for all three modes. Figure 3b shows the integrated average RIN values and the difference of the average RIN from the average value (−133.54 dB/Hz) for 19 selected modes across the wavelength range from 1543 nm to 1556 nm (which contains a total of 56 modes). It is observed that the average RIN is below −132.2 dB/Hz, and the difference is <1.3 dB for the modes between 1543 and 1555 nm. At the long wavelength range close to 1556 nm, the average RIN and the difference are increased to −128.7 dB/Hz and 4.8 dB. However, the measured RIN values of the all-longitudinal modes are below the IEEE standard (−128 dB/Hz) for RIN with Terabit/s communications as demonstrated in [17].

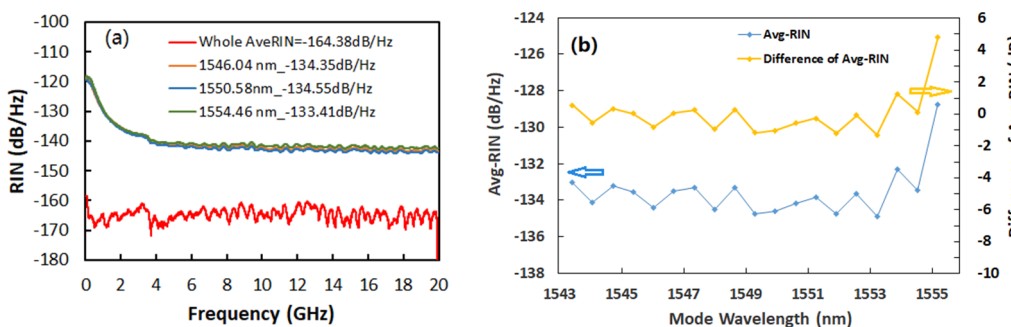

**Figure 3.** (**a**) RIN spectra including all the modes (red) and three filtered individual channels at 1546.04 nm (brown), 1550.58 nm (blue), and 1554.46 nm (green) for the laser measured at an injection current of 400 mA and 18 °C in the frequent range from 10 MHz to 20 GHz; (**b**) calculated average RIN (blue) over the 20 GHz frequency bandwidth from the RIN spectra and the difference (yellow) of average RIN from their average value (−133.54 dB/Hz) for 19 filtered individual modes in the wavelength range from 1543 to 1556 nm.

### 3.3. Optical Linewidth (Phase Noise)

Phase noise of OFC sources is becoming a very critical parameter for higher-order QAM modulation formats [38]. Phase noise normally is presented by optical mode linewidth. Figure 4a shows the measured phase noise for the selection of individual

longitudinal lasing modes in the wavelength range from 1543 to 1556 nm. The laser was driven at 400 mA at a temperature of 18 °C. The phase noise is obtained by analyzing the frequency noise spectra using an optical auto-correlator. A narrowband optical tunable filter (EXFO XTM-50) is used for filtering the individual longitudinal lasing modes. The minimum measured phase noise is 0.19 MHz at a wavelength of 1552.4 nm, and all values of phase noise are below 1.5 MHz. The phase noise as a function of mode number is fitted to a parabolic curve, as shown in Figure 4b. A technique for estimating the RF linewidth ($\Delta f_{FR1}$) from the parabolic curve fitting of phase noise ($\Delta \upsilon_n$) vs. mode number ($\Delta n$) for a SSP MLL has been recently studied [10]. Using the deduced explicit expression as shown in [10] Equation (7), parabolic fitting results of a minimum linewidth of $\Delta \upsilon_{min}$ = 240 kHz at the 13th mode is extracted for the 28.4 GHz laser with QDot gain material. The first harmonic RF linewidth is then extracted from the parabolic fit giving $\Delta \upsilon_{RF1}$ = 0.631 kHz.

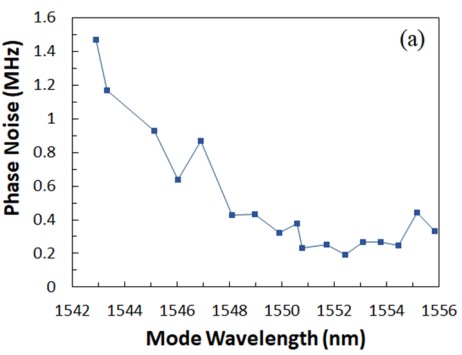 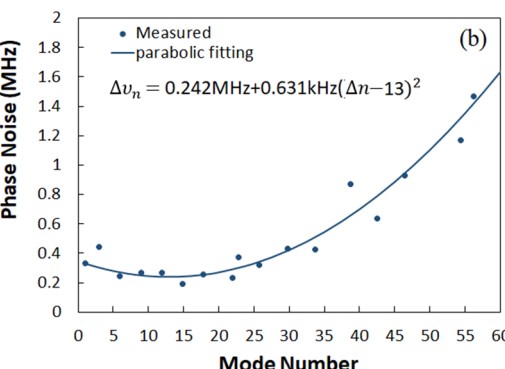

**Figure 4.** Phase noise vs. mode lasing wavelength (**a**), and mode number with parabolic fit (**b**), measured using an OEWaves OE4000 automated laser linewidth/phase noise measurement system. The laser was driven at 400 mA at a temperature of 18 °C.

### 3.4. RF Beating Note and Timing Jitter

To verify the $\Delta \upsilon_{RF1}$ obtained using the optical technique described in the above section, we performed a direct measurement of the RF linewidth using a high-*speed* photodetector. We focused all modes of the laser onto a 45 GHz IR photodetector and monitored the electrical output using a 50 GHz PXA signal analyzer. The RF signal analyzer was set to a 51 Hz resolution bandwidth (RBW) and 100 kHz span. Figure 5a shows the measured normalized first harmonic RF power spectral density (PSD) and Lorentzian curve fit for the laser with the frequency offset for clarity. A peak frequency $f_0$ of 28.427 GHz was obtained, where $f_0$ represents the pulse repetition rate, which is consistent with the measured mode spacing from the optical spectrum. The Lorentzian line shape provide good fit for the measured RF PSD curve. The extracted FWHM Lorentzian linewidths, $\Delta f_{FR1}$, of 0.654 kHz is obtained. Since the properties of intrinsic phase noise from relatively broadband spontaneous emission in the passively ML laser leads to a Lorenzian-shaped PSD of photocurrent RF phase noise, an expression of RMS pulse-to-pulse time jitter estimated from a Lorenzian fitted RF linewidth was proposed in [18] Equation (3). The pulse-to-pulse timing jitter of 2.13 and 2.09 fs are obtained from the direct RF linewidth measurement ($\Delta \upsilon_{RF1}$ = 0.654 kHz) and the parabolic curve fit of phase noise vs. mode number through RF linewidth ($\Delta \upsilon_{RF1}$ = 0.631 kHz), respectively.

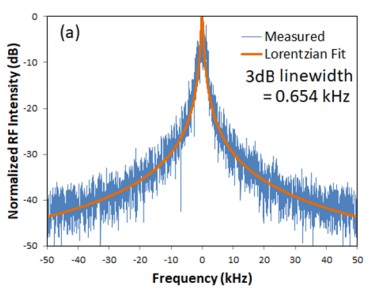
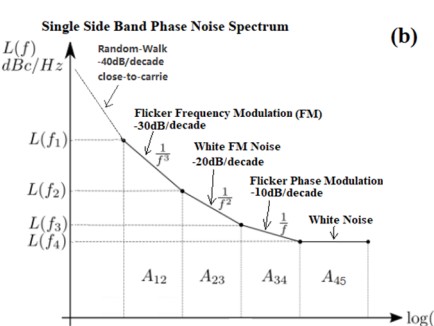
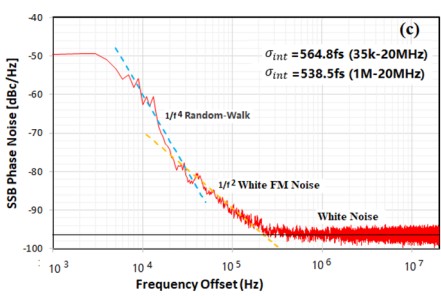

**Figure 5.** (**a**) Normalized first harmonic RF PSD with Lorentzian fit for the laser measured using a Keysight Technologies N9030A 50 GHz PXA signal analyzer at RBW = 51 Hz. (**b**) A typical single side band (SSB) phase noise spectrum L(f) showing the regions of the different noise mechanisms with various slopes, frequency boundaries, and areas of the noise mechanism regions. (**c**) Calculated SSB L(f) from the laser showing the two main noise mechanisms from measured RF PSD at RBW = 10 kHz.

In the case of a semiconductor passively mode-locked laser, it is shown that due to material nonlinearities and saturated absorber in the gain area, the phase correlation of the optical longitudinal modes can take place resulting in a strong four wave mixing [39]. This is especially true for an active region consisting of QDot or QDash structures. The common pulse to clock RMS time jitter, also called integrated timing jitter, can be obtained from integration of the single side band (SSB) phase noise spectrum *L(f)* by [40,41]

$$\sigma_{int} = \frac{1}{2\pi f_0}\sqrt{2\int_{f_{low}}^{f_{high}}L(f)df} = \frac{1}{2\pi f_0}\sqrt{2(A_{12}+A_{23}+A_{34}+A_{45})} \qquad (1)$$

where $f_{high}$ and $f_{low}$ are the upper and lower integration frequencies, respectively; *L(f)* is equal to the normalized RF PSD for a one Hz bandwidth in the interested range; and $A_{12}$, $A_{23}$, $A_{34}$, and $A_{45}$ are corresponding to the areas of $1/f^3$ flicker frequency modulation (FFM), $1/f^2$ white flicker frequency noise (WFF), $1/f^1$ flicker phase modulation (FPM), and $1/f^0$ white (shot) noise regions, respectively [40]. A typical *L(f)* for a passively MLL is shown in Figure 5b where $f_1$, $f_2$, $f_3$, and $f_4$ indicate the frequency boundaries of the different noise mechanisms with corresponded slopes. As pointed out in [41], this phase noise estimation method is only valid for passively MLLs at frequency offsets well above the carrier peak linewidth.

Figure 5c shows measured SSB *L(f)* which is obtained from the normalized measured RF PSD for a one Hz bandwidth in the offset frequency range from center (as origin) to 20 MHz at the RBW setting of 10 kHz for the laser. From the colored dashed lines with negative slopes corresponding to the different noise mechanisms, it is observed that $1/f^2$ WFF noise is dominated at the low offset frequency from 35 kHz to 250 kHz, which is the common type of noise found in a passively mode-locked laser. It intersects $1/f^4$ random walk at the phase noise level of −79.8 dBc/Hz, which is caused by external environment effects on the laser. The random-walk noise near carrier peak contributes to the laser's pulse-to pulse timing jitter. $1/f^3$ FFM and $1/f$ FPM noises are total masked. Above 250 kHz, the frequency independent shot noise dominates at level of −96.5 dBc/Hz. The integrated timing jitter of 564.8 fs is obtained by using Equation (1) in the frequency range from 35 kHz to 20 MHz, which is contributed to by $1/f^2$ WFF and shot noises. Above 20 MHz, the phase noise is affected by the noise floor level of the ESA, thus limiting the contribution of the high frequency components (>20 MHz) to the timing jitter as described in [42].

## 4. Discussions

The performance of coherent networks highly depends on the properties of optical source, such as the number and power of comb modes, OSNR, RIN, phase noise, and timing jitter [43]. The laser investigated in this paper possesses related high individual optical

powers (over −16.6 dBm) for all 65 modes as shown in Figure 2a. In addition, the extremely high OSNR (>60 dB) with the central wavelength located at the middle of the C-band are benefits for achieving high performance in a multi-channel parallel transmission WDM with high-order advanced modulation formats [44].

The all-modes integrated average RIN of −164.4 dB/Hz achieved for our QDot SSP MLL is much lower than that (−140 dB/Hz) from the QDash SSP MLL reported in Ref [45]. On the other hand, RINs of individual single longitudinal modes are important parameters since each of the modes of the laser are to be used as transmission channels. They are expected to have a higher RIN in comparison to the collective RIN of all the modes due to mode partition noise, which arises due to an intensity anti-correlation between the longitudinal modes [46,47]. The small values of the average RIN between −134.8 and −128.7 dB/Hz for all 56 modes obtained in this study are attributed to the characteristics of QDot material inside the laser cavity with very low mode partition noise.

According to the analysis of tolerance against the laser linewidth ($\Delta v$) times symbol duration ($T_s$) product ($\Delta v \cdot T_s$) for a receiver penalty of 1 dB at a BER of $10^{-3}$ [48], the maximun tolerable linewidth for a 28 Gbaud system are 3.92 MHz and 1.12 MHz for the square 16 QAM and 64 QAM modulation formats, respectively. From the measured results shown in Figure 4, all 56 modes of our laser are suitable for the 28 Gbaud 16 QAM system and most modes are suitable for the 28 Gbaud 64 QAM system.

The value of the RF linewidth of a semiconductor MLL is strongly affected by the gain material dimensionality. It decreases when moving from bulk to QW and from QW to QDot/Qdash active regions, as shown in Figure 6. Typical values of the RF linewidth lie in the range of a few tens of kilohertz [49] and a few kilohertz [10] for QDash active regions down to a few hundreds of hertz reported in this study for QDot MLLs, compared with the much larger RF linewidth for bulk and QW MLLs.

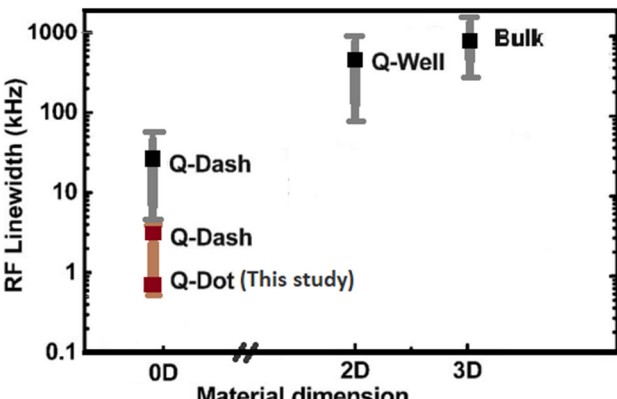

**Figure 6.** RF linewidth dependence on gain material dimensionality, where Bulk, Q-Well, and Q-Dash (black) are from [49], Q-Dash (brown) from [10].

Study of the RF linewidth either from a directly measurement of the RF spectrum or curve fitting of measured phase noise vs. mode number can provide simple and appropriate ways to characterize the pulse-to-pulse timing jitter of a semiconductor passively ML laser. The results from the two methods presented in this work are in excellent agreement. The method of parabola curve fitting from measured phase noise vs. mode number does not require a direct measurement of the RF PSD, which for a high repetition rate laser necessitates the use of a high-speed photodetector. Therefore, this method is not restricted to measuring lasers with repetition rates below the bandwidth of the used photodetector. The femtosecond level pulse-to-pulse timing jitter obtained from the laser studied in this study demonstrates highly stable mode spacing across the whole comb.

An integrated timing jitter of 790 fs is reported in the frequency range from 1 to 20 MHz for a 40 GHz InAs/InP QDash SSP MLLs [42]. In order to make a comparison, we

calculated an integrated timing jitter of 538.5 fs in the same frequency range (1–20 MHz) for our laser with a QDot gain material, a significant improvement over that reported in [42].

## 5. Application

Figure 7 shows the experimental setup for the dual-polarization QAM data format transmission. A 28 GBd 16 QAM base-band signal is created by using an arbitrary waveform generator (AWG, 65 GSa/s, 25 GHz bandwidth) with a pseudo-random bit sequence with a pattern length of $2^{15}$-1 on four channels (IX/IY/QX/QY). A root-raised-cosine filter is applied with the roll-factor of 0.35 for Nyquist pulse filtering. Thermally stable nested double polarization lithium niobate Mach-Zehnder modulators that constitute two *I/Q* optical modulators are driven by the AWG for data transmission. The encoded optical signal is transmitted for both back-to-back (B2B) configuration and over 100 km SSMF, respectively. The encoded signal is amplified by an erbium doped fiber amplifier (EDFA) and then an optical band pass filter is used to filter out the amplified spontaneous emission from the EDFA. At the receiver side, an optical modulation analyzer (OMA, 63 GSa/s, 23 GHz bandwidth) coherently receives the signals using a free-running local oscillator. Combined with the vector signal analyzer software package offered by the OMA, the sampled waveforms are processed with front-end correction and de-skewing, then compensated for chromatic dispersion and frequency offset. Afterwards, the signals are match-filtered and synchronized for time-domain equalization. Finally, the output 16 QAM signal is decoded for error-vector-magnitude (EVM) measurement and BER evaluation.

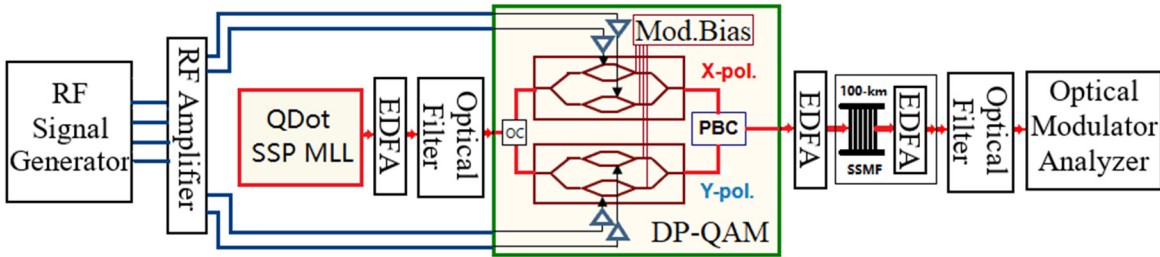

**Figure 7.** Setup schematic of a dual polarization QAM data format transmission system. The laser studied in this paper is used as the coherent transmitter source. All other equipment used is listed in Table 1.

**Table 1.** The main equipment used in our DP-QAM format data transmission system.

| Equipment | Vendor | Module |
|---|---|---|
| RF Signal Generator | Keysight | M8195A |
| RF Amplifier | Centellax | OA3MHQM |
| EDFA | Amonics | AEDFA-PA-35-B-FA |
| Optical Filter | Santec | OTF-350 |
| DP-QPSK Modulator | Fujitsu | FTM7977HQA |
| Modulator Bias Controller | ID Photonics | MBX |
| Optical Modulator Analyzer | Keysight | N4392A |

Figure 8a shows the BER performance of each channel at the same received optical power of −10 dBm for 16 randomly selected channels in the B2B system and all individual channels after inserting 100 km SSMF. All BER values are exhibited below the hard-decision FEC (HD-FEC) limit (BER = $3.8 \times 10^{-3}$) [50] with B2B and after 100 km SSMF transmission.

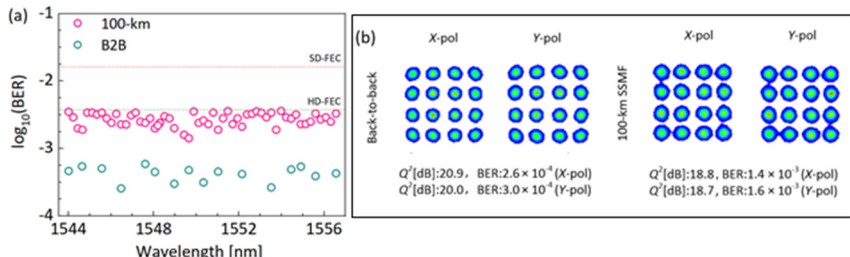

**Figure 8.** (**a**) BER evaluation over the mode wavelength range for B2B and after 100 km of SSMF transmission. All the selected channels are below the SD-FEC limit (BER = $2.0 \times 10^{-2}$) and HD-FEC limit (BER = $3.8 \times 10^{-3}$). (**b**) Constellation diagram obtained for a selected channel at 1552.180 nm for both *X*- and *Y*-polarization after B2B and 100 km of SSMF transmission. The color indicates the relative density of symbols detected in the complex plane, with red indicating a higher density and blue a lower density.

Figure 8b shows constellation diagrams for the channel at 1552.180 nm wavelength with B2B and after 100 km SSMF transmission. In the B2B configuration, a signal quality ($Q^2$[dB] $\approx 20log_{10}\left(\frac{1}{EVM}\right)$) [44] higher than 20 dB is achieved. It drops to 18.7 dB when adding 100 km SSMF, which is probably due to the dispersion in the SSMF. The achieved aggregate line-rate data transmission capacity of the system is 12.5 Terabit/s (16QAM 56 $\times$ 28 GBd PDM).

## 6. Conclusions

The performance of an InAs/InP QDot SSP MLL with a mode spacing of 28.4 GHz is investigated in detail. For the 56 filtered individual channels, the average RIN value is –133.54 dB/Hz and the phase noise is less than 1.5 MHz for each individual channel. Moreover, a pulse-to-pulse timing jitter of 2.1 fs and integrated timing jitter of 564.8 fs in the frequency range from 35 kHz to 20 MHz are obtained. It is observed that the $1/f^2$ white flicker frequency is the main contribution to the phase noise in the low frequency (<250 kHz), and shot noise dominants when frequency is >250 kHz. By employing this ultralow noise and timing jitter laser with 56 wavelength channels as optical carriers, 12.5 Terabit/s aggregate data transmission capacity is demonstrated with dual-polarization 16 QAM and base modulation rate of 28 GBd over a 100 km standard single-mode fiber. This performance is obtained without any form of feedback to control the laser linewidth. These results may lead to an important step towards small size, cost-efficient, low noise, low timing jitter, and a high number of multi-channel light sources for optical networking systems with the capacity of Terabit/s or even higher.

**Author Contributions:** Conceptualization, Y.M., Z.L. and J.L.; methodology, Y.M., G.L., K.Z., Z.L., J.L., P.J.P., C.-Y.S. and P.B.; software, G.L. and K.Z.; validation, Z.L., J.L. and P.J.P.; formal analysis, Y.M, G.L., K.Z. and C.-Y.S.; investigation, Y.M., G.L., K.Z., Z.L., J.L., P.J.P., C.-Y.S. and P.B.; resources, Z.L.; data curation, Y.M, G.L., K.Z. and C.-Y.S.; writing—original draft preparation, Y.M.; writing—review and editing, Y.M., G.L., K.Z., Z.L., J.L., P.J.P., C.-Y.S. and P.B.; visualization, Y.M., P.J.P., P.B. and K.Z.; supervision, Z.L., J.L., and P.J.P.; project administration, Z.L. and J.L.; funding acquisition, Z.L. and J.L.; All authors have read and agreed to the published version of the manuscript.

**Funding:** This research project is supported by the National Research Council Canada's high-throughput and secure network (HTSN) program.

**Institutional Review Board Statement:** Not applicable.

**Informed Consent Statement:** Not applicable.

**Data Availability Statement:** Not applicable.

**Conflicts of Interest:** The authors declare no conflict of interest.

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
