# Peer review of "Ultralow Noise and Timing Jitter Semiconductor Quantum-Dot Passively Mode-Locked Laser for Terabit/s Optical Networks"

_photonics, doi:10.3390/photonics9100695_

Round 1

Reviewer 1 Report

The authors have systematically investigated the performance of quantum dot single-section passively mode-locked lasers especially for the phase noise and timing jitter. The paper is well organized, and experiments are well-designed. I think the paper should be published after correcting the following two points.

1.      Please remove the red wavey underlines in Table 1.

2.      There are many acronyms used in this paper, but some of them are not spelled out or not spelled out for the first time when they’re used. For example, ‘RF’, ‘SEM’,’BER’, ‘B2B’, ‘EDFA’, ‘ASE’ is not spelled out throughout the paper; ‘PSD’ appears in L215 the first time, but it is not spelled out; ‘dual-polarization’ and ‘quantum dots’ sometimes are written as ‘DP’ and ‘QDots’ and sometimes are written in its full term; ‘Fabry-Perot’ has been shown as ‘F-P’ twice in this paper; Abbreviations like ‘AlN’, ‘Au’, ‘AuSn’, ‘TEC’, ‘PRBS’, ‘RRC’, ‘LiNbO3’, ‘OBPF’, ‘LO’ and ‘EVM’ are only used once, so there’s no need to show the abbreviations for these words. Please check again the whole paper for inconsistency.

Author Response

Response to Reviewer 1 Comments

Point 1: Please remove the red wavey underlines in Table 1.

Response 1: The red wavey underlines in Table1 was removed.

Point 2: There are many acronyms used in this paper, but some of them are not spelled out or not spelled out for the first time when they’re used. For example, ‘RF’, ‘SEM’,’BER’, ‘B2B’, ‘EDFA’, ‘ASE’ is not spelled out throughout the paper; ‘PSD’ appears in L215 the first time, but it is not spelled out; ‘dual-polarization’ and ‘quantum dots’ sometimes are written as ‘DP’ and ‘QDots’ and sometimes are written in its full term; ‘Fabry-Perot’ has been shown as ‘F-P’ twice in this paper; Abbreviations like ‘AlN’, ‘Au’, ‘AuSn’, ‘TEC’, ‘PRBS’, ‘RRC’, ‘LiNbO3’, ‘OBPF’, ‘LO’ and ‘EVM’ are only used once, so there’s no need to show the abbreviations for these words. Please check again the whole paper for inconsistency.

Response 2:  Thanks for pointing out these acronyms issues. The ‘RF’, ‘SEM’,’BER’, ‘B2B’, ‘EDFA’, ‘ASE’, and ‘PSD’ are spelled at the 1st appear. ‘dual-polarization’ and ‘quantum dots’ are unified. ‘AlN’, ‘Au’, ‘AuSn’, ‘TEC’, ‘PRBS’, ‘RRC’, ‘LiNbO3’, ‘OBPF’, and ‘LO’ are moved out. Only EVM is remained because it is shown in this sentence: “In the B2B configuration, a signal quality (Q2[dB] ≈  ) [40] higher than 20 dB is achieved”. Please see the following revisions in details:

In Abstract (page 1): changed “WDM” to “wavelength division multiplexing”; “dual-polarization (DP)” to “dual-polarization”.

In Introduction:

1st paragraph(page 1): changed “DFB laser” to “distributed-feedback laser”.

2nd paragraph (page 1-2): changed “quantum dots or dashes” to “quantum dots (QDots) or quantum dashes (QDashes)”; “quantum dot or dash” to “QDot or QDash”; “RF” to “radio frequency (RF)”.

3rd paragraph (page 2): changed “SSMF” to “standard single-mode fiber (SSMF)”; BER to bit error rate (BER); “Qdots vs. QDashs” to “Qdot vs. QDash”.

In Materials and Methods:

1st paragraph (page 2-3): changed “SEM” to “scanning electron microscope (SEM)”; “Fabry-Perot (F-P)” to “F-P”; “chemical beam epitaxy (CBE)” to “chemical beam epitaxy”; “photoluminescence (PL)” to “photoluminescence”; “metal organic chemical vapour deposition (MOCVD)” to “metal organic chemical vapour deposition”.

In Figure 1 captain (page 3): changed “scanning electron microscopy (SEM)” to “SEM”; “Fabry-Perot” to “F-P”.

2nd paragraph (page 3): changed “Fabry-Perot” to “F-P”.

3rd paragraph (page 3): changed “Aluminum Nitride (AlN)” to “Aluminum Nitride”; “gold (Au)” to “gold”; “Eutectic Gold Tin (AuSn)” to “Eutectic Gold Tin”; “thermoelectric cooler (TEC)” to “thermoelectric cooler”; “laser diode driver controller (LDC)” to “laser diode driver controller”.

In Experimental Results:

3.2. Title changed from “Relative Intensity Noise (RIN)” to “RIN”:

1st paragraph (page 4): changed “relative intensity noise (RIN)” to “RIN”

3.4. RF Beating Note and Timing Jitter:

1st paragraph (page 6): changed “PSD” to “power spectrum density (PSD)”

2nd paragraph (page 6): changed“single side band” to “single side band (SSB)”.

In Application:

1st paragraph (page 9): changed “pseudo-random bit sequence (PRBS)” to “pseudo-random bit sequence”; “root-raised-cosine (RRC)” to “root-raised-cosine”; “lithium niobate (LiNbO3)” to “lithium niobate”; “B2B” to “back-to-back (B2B)”; “EDFA” to “erbium doped fiber amplifier (EDFA)”; “ band pass filter (OBPF)” to “optical band pass filter”; “ASE” to “amplified spontaneous emission”; “local oscillator (LO)” to “local oscillator”.

In Conclusions (page 10): changed “DP” to “dual-polarization”.

Reviewer 2 Report

Diode optical frequency comb lasers are hot topics, and this laser is a promising multi-channel optical sources for WDM transmission systems with advanced modulation formats. As described by the authors, they “have reported InAs/InP QDot and QDash-MLLs with channel spacings from GHz to 65 THz [18, 23-25, 27-32]”. This article investigates the detailed properties of a QDot based InAs/InP SSP MLL, especially the relative intensity noise (RIN), phase noise, pulse-to pulse, and pulse-to-clock (integrated) timing jitter. Furthermore, by use of the InAs/InP QDot SSP MLL, the authors demonstrated a 12.5 terabit/s optical network over 100-km of SSMF with 16-QAM modulation. It is a nice work, and this investigation will benefit to the field of optical communications. I recommend it can be accepted.

Some suggestions:

(1)    Please give the full spell of “QDot” and “QDash” (probably in the 41th line).

(2)    Table 1: The name of 5 vendors (the 2nd column) are highlighted. Please confirm it.

(3)    56 lasing modes were available for this laser, and the noise and timing jitter is small. What is the limitation of the number of comb with relatively large power? What is the possible limitation of the noise and timing jitter?

Author Response

Response to Reviewer 2 Comments

Point 1: Please give the full spell of “QDot” and “QDash” (probably in the 41th line)

Response 1: In the 2nd paragraph (page 1, 41th line) in Introduction: changed “quantum dots or dashes” to “quantum dots (QDots) or quantum dashes (QDashes)”.

Point 2: Table 1: The name of 5 vendors (the 2nd column) are highlighted. Please confirm it.

Response 2: The highlighted name of 5 vendors (red wavey underlines) in Table1 was removed.

Point 3: 56 lasing modes were available for this laser, and the noise and timing jitter is small. What is the limitation of the number of comb with relatively large power? What is the possible limitation of the noise and timing jitter?

Response 3:

1, what is the limitation of the number of comb with relatively large power?

There are possible two main limitations of the number of comb with relatively large power:

  1. a) Generally, the effects of the inhomogeneous dot size and composition distribution in the active gain region, and the processing conditions during the material growth would contribute to the broadening of the gain. A balance between the inhomogeneous dot size and the laser power density have to be considered in order to obtain more number of comb with relatively large power. In our observation, the PL FWHM at low temperature is quite high (in the order of 80 nm) meaning that the size distribution is large enough to provide a very broadband gain. When lasing, the FWHM is reduced dramatically to around 12 nm which is significantly smaller than the potential gain bandwidth. It is possible that there is some redistribution of carriers among the different size dots which creates an effective narrowing of the gain when lasing.
  2. b) Limitation of the dispersion in the cavity for mode-locked lasers. We are working on dispersion compensation for fully exploring the capability of the QDot MLL.

2, what is the possible limitation of the noise and timing jitter?

The main effects for broadening the linewidth and increasing timing jitter for the QD MLL are:

1, fundamentally, it is caused by the coupling of spontaneous emission into the oscillating mode, leading to white noise.

2, effects such as flicker frequency noise occur at longer time scales, i.e., lower frequencies.

3, effect of random-walks caused from external environment influences, such as temperature fluctuations and mechanical vibrations, happen on an even larger time scale.

From the measurement results shown in the paper, the main limitation of pule-to-pulse timing jitter which is proportional to RF FWHM linewidth is probably the random-walk noises, while the main limitation of pule-to-clock timing jitter is probably the flicker frequency noise and white noise for the QD SSP MLL.